

# Exogenous applications of brassinosteroids promote secondary xylem differentiation in *Eucalyptus grandis*

Fangping Zhou[1,2,*], Bing Hu[1,*], Juan Li[1], Huifang Yan[3], Qianyu Liu[4], Bingshan Zeng[1] and Chunjie Fan[1,2]

[1] Key Laboratory of State Forestry Administration on Tropical Forestry, Research Institute of Tropical Forestry, Chinese Academy of Forestry, Guangzhou, China
[2] State Key Laboratory of Tree Genetics and Breeding, Chinese Academy of Forestry, Beijing, China
[3] School of Life Sciences Fudan University, Shanghai, China
[4] State Key Laboratory for Conservation and Utilization of Subtropical Agro-bioresources, South China Agricultural University, Guangzhou, China
* These authors contributed equally to this work.

Corresponding author
Chunjie Fan, fanchunjie@caf.ac.cn

## ABSTRACT

Brassinosteroids (BRs) play many pivotal roles in plant growth and development, especially in cell elongation and vascular development. Although its biosynthetic and signal transduction pathway have been well characterized in model plants, their biological roles in *Eucalyptus grandis*, a major hardwood tree providing fiber and energy worldwide, remain unclear. Here, we treated *E. grandis* plantlets with 24-epibrassinolide (EBL), the most active BR and/or BR biosynthesis inhibitor brassinazole. We recorded the plant growth and analyzed the cell structure of the root and stem with histochemical methods; then, we performed a secondary growth, BR synthesis, and signaling-related gene expression analysis. The results showed that the BRs dramatically increased the shoot length and diameter, and the exogenous BR increased the xylem area of the stem and root. In this process, EgrBRI1, EgrBZR1, and EgrBZR2 expression were induced by the BR treatment, and the expressions of HD-ZIPIII and cellulose synthase genes were also altered. To further verify the effect of BRs in secondary xylem development in Eucalyptus, we used six-month-old plants as the material and directly applied EBL to the xylem and cambium of the vertical stems. The xylem area, fiber cell length, and cell numbers showed considerable increases. Several key BR-signaling genes, secondary xylem development-related transcription factor genes, and cellulose and lignin biosynthetic genes were also considerably altered. Thus, BR had regulatory roles in secondary xylem development and differentiation via the BR-signaling pathway in this woody plant.

## INTRODUCTION

Brassinosteroids (BRs), plant steroid hormones, play a critical role in plant cell division and elongation, photomorphogenesis, vascular differentiation, and responding to biotic
and abiotic stress (*Rao & Dixon, 2017*; *Shahzad et al., 2018*; *Wei & Li, 2016*; *Zhu, Sae-Seaw & Wang, 2013*; *Kumar, Campbell & Turner, 2016*; *Shahzad et al., 2018*). Researchers have extensively documented the metabolism and perception of BRs (*Belkhadir & Jaillais, 2015*; *Clouse, 2011*; *Yang et al., 2011*). In plants, BR synthesis mainly includes several key processes. Firstly, campesterol is converted to campestanol (CN) by de-etiolated 2 (DET2). Then, several cytochrome P450 enzymes such as Dwarf4 (DWF4), constitutive photomorphogenesis and dwarfism (CPD), Rotundifolia3, CYP90D1, and Brassinosteroid-6-oxidase 1 (BR6ox1) successively act and form castasterone (CS) *via* the different C-6 oxidation pathway. Finally, CS is transformed into brassinolide (BL) under BR6ox2 acting (*Wei & Li, 2016*). In addition, researchers recently identified a CN-independent pathway of BR biosynthesis, in which DWF4 is considered to be the main rate-limiting factor (*Ohnishi et al., 2012*; *Wei & Li, 2016*). Combining genetics, biochemistry, and molecular biology, researchers have also established the BR-signaling transducing pathway in the model plant *Arabidopsis* and rice. The BRs are firstly perceived by the main receptor kinase Brassinosteroid-insensitive 1 (BRI1) and trigger an intracellular signal transduction cascade. In this process, BRI1 activates and recruits its coreceptor kinase Bri1-associated receptor kinase1 (BAK1) while disassociating the inhibitory protein BRI1 kinase inhibitor1 (BKI1) *via* transphosphorylation. The activated BRI1 can phosphorylate BR SIGNALING KINASE (BSK1), and subsequently BSK1 activates BRI1 SUPPRESSOR1 (BSU1). BSU1 inactivates Brassinosteroid insensitive2 (BIN2), which phosphorylates Brassinazole resistant1 (BZR1) and BZR2 and BES1 (BRI1-EMS-Suppressor1) and induces them to remain in a cytoplasmic interaction with 14-3-3 proteins. Dephosphorylated BIN2 is then degraded by the proteasome. Finally, dephosphorylated BZR1 and BZR2 move into the nucleus where they directly control the expression of BR target genes (*Belkhadir & Jaillais, 2015*; *Kim & Wang, 2010*; *Zhu, Sae-Seaw & Wang, 2013*).

BR functions in herbaceous plants and are involved in the primary cell wall in plant development (*Xie, Yang & Wang, 2011*) and cellulose biosynthesis (*Sánchez-Rodríguez et al., 2017*). Thus, several researchers have focused on tracheary element (TE) differentiation, cell wall lignification, xylem vessel formation, and differentiation (*Lee et al., 2021*; *Nolan et al., 2020*; *Xie, Yang & Wang, 2011*; *Yamamoto, Demura & Fukuda, 1997*; *Yamamoto et al., 2001*; *Yamamoto et al., 2007*). Recently, researchers have also explored BR functioned in secondary growth and wood formation. In *Liriodendron tulipifera*, applied BR *in vivo* could modify secondary cell wall assembly and components (*Jin et al., 2014*). In *Populus*, several groups verified that the exogenous application of brassinosteroid increased in secondary growth by promoting procambial cell division and secondary xylem differentiation and stimulating secondary xylem fiber elongation (*Du et al., 2020*; *Gao et al., 2019*; *Wang et al., 2022*; *Yuan et al., 2019*). Moreover, the overexpression of the BR biosynthesis gene BR6OX and BR main receptor kinase homolog gene PtBRI1.2 could enhance xylem formation and cell wall thickness (*Jiang et al., 2021*; *Jin et al., 2020*; *Jin et al., 2017*). Similarly, the exogenous application of 24-epibrassinolide could alter xylan content in the xylem fibers and increase the syringyl lignin content of cell walls in the xylem (*Pramod et al., 2021*). Moreover, overexpression of key enzyme genes for BR synthesis also promotes xylem development, especially secondary cell wall formation. It was found that when *PtoDET2*

and *PtoDWF1*, the key enzymes for BR synthesis genes (*Fan et al., 2020*), these two genes were strongly expressed in the secondary cell wall formation site of the stem, and the xylem area of the transgenic plants increased, and the stem diameter and the number of xylem cell layers increased significantly. Besides, studies in poplar showed that BR can induce upregulation of NAC transcription factors *PtNAC1* and *PtNAC104*, and subsequently regulate xylem development, overexpression of *PtBRI1.2* in poplar can also promote xylem development (*Jiang et al., 2021*). These results indicated that BRs have an important role in secondary xylem formation and development in woody plants.

*Eucalyptus* spp. have increasingly become one of the most important sources of timber and pulp in the world because their rapid growth, wide adaptation, and superior wood property. The completion of the *E. grandis* genome provides possibilities to elucidate the molecular regulating mechanism of its wood formation (*Myburg et al., 2014*). Unfortunately, the effects of BRs acting in Eucalyptus growth, especially in wood formations, are poorly understood. This work mainly aims to verify the effects of exogenous BRs in *E. grandis*, with special emphasis on the relationship between exogenous BRs and xylem development. Meanwhile, an exhaustive morphological and anatomical characterization was performed. Moreover, the transcription of the genes involved in BR homeostasis, signal transduction, and secondary xylem development was also performed to further explain the effects of exogenously applied BRs. This study will provide novel insight into the role of BR-mediated secondary xylem development and wood formation in *Eucalyptus*.

## MATERIAL AND METHODS

### Plant material and exogenously applied hormone

Plantlets of *E. grandis* clone GL1 were used as plant material. The propagation buds were grown in a culture room. Selected robust buds were cultured for 7 days on root induction medium consisting of modified 1/2 MS medium with 3.0% sucrose and 5.0 g $L^{-1}$ phytogel supplemented with 0.20 mg $L^{-1}$ 6-benzyladenine and 0.05 mg $L^{-1}$ a-naphthalene acetic acid (*Liu et al., 2018*). To evaluate the effect of treatments of 24-epibrassinolide (EBL) and/or brassinazole (BRZ, an inhibitor of BR biosynthesis), rooted plantlets (with ~0.5 cm root length) in glass jars were subsequently moved to 250 mL glass jars containing 30 mL of 1/2 MS medium of pH 5.8 containing EBL (1.0 μg $L^{-1}$), BRZ (1.0 mg $L^{-1}$), or the combination of both EBL and BRZ. All treatments were performed for three weeks and three times at different periods. Each jar contained five explants, with four jars per replicate and three replicates per treatment. Each experiment was carried out three times at different periods. All cultures were maintained in a culture room at $25 \pm 1\,°C$ with 16-h photoperiod and light intensity of 80 μmol $m^{-2}$ $s^{-1}$ provided by cool-white fluorescent lights.

To determine whether BR affects growth promotion and xylem development in *E. grandis*, EBL was applied directly to secondary xylem of 6-month-old *E. grandis* GL1 plant stems after debarking. Each tree planted in an individual pot containing turf soil was grown in glasshouse under natural light conditions at the Research Institution of Tropical

Forestry, Chinese Academy of Forestry (23°19′N, 113°38′E). Forty plants with similar stem diameters and heights (80–90 cm) were selected for EBL treatment. Of lanolin paste containing EBL (1.0 ng mL$^{-1}$), one mL was applied once to a debarked stem area in the sixteenth internode from the shoot apex (*Jin et al., 2014*), and lanolin alone was applied as a control (Fig. S1).

## Anatomical structure analysis

To examine the effects of EBL or BRZ treatment on Eucalyptus growth, anatomical structure of the stem and root was investigated. Segments of basal stem (about 1.0 cm in length) and root in foregoing EBL or/and BRZ treatments were harvested. Then samples were cut into small pieces of approximately 2-3 mm and embedded in 5.0% agar. Cross-sections of thickness 50 μm were cut using a vibrating blade microtome (Leica VT1000S, Heidelberg, Germany). For visualization of xylem cells, sections were stained briefly with 0.1% toluidine blue O (TBO) solution for 10 s, mounted in 50% glycerol, and observed under a BX43 light microscope (Olympus, Tokyo, Japan). Area of xylem and cross-section of stem and area of xylem and stele in root were measured by ImageJ (*Schneider, Rasband & Eliceiri, 2012*).

To examine the effects of EBL treatment on Eucalyptus stem, small pieces (approximately two mm × 4 mm × 6 mm) from the treatment site were carefully harvested with a razor blade and immediately fixed in FAA solution containing 5.0% glycerine for further histochemical analysis (*Jin et al., 2014*). They were then infiltrated and embedded with paraffin, and sections (thickness 10 μm) were obtained and stained with TBO. The stained slides were analyzed and imaged by light microscopy with a BX21 digital camera (Olympus, Tokyo, Japan).

## Measuring lengths of fiber cells

The samples from the treatment site, similarly to the previous histochemical analysis, were harvested and specimens immediately soaked and stored in Schulze's reagent (6% (w/v) KClO$_3$ in 50% (v/v) nitric acid) for 1 week at room temperature. The specimens were heated at 60 °C for 30 min by thermostatic water bath, washed three times with sterile and distilled water, and shaken vigorously to dissociate fiber cells and vessel elements. Dissociated fiber cells and vessel elements in water were analyzed and photographed using light microscopy. The lengths of more than 100 individual cells in each sample were respectively measured for fiber and vessel elements.

## RNA extraction, reverse transcription, and real-time quantitative PCR analysis

An EASYspin Plus RNA kit (Aidlab, Gdansk, Poland) was used for total RNA isolation and RNase-free DNase I (Qiagen, Hilden, Germany) was applied for removing genomic DNA. SuperScript III (Invitrogen, Waltham, MA, USA) applied for cDNA synthesis, which was used for further qRT-PCR. The experiment was performed by using SYBR kit (TaKaRa Biotechnology, Shiga, Japan) and LightCycle 96 real-time PCR machine (Roche, Basel, Switzerland). The detailed process and data analysis were developed as described previously by *De Oliveira et al. (2012)* and *Liu et al. (2018)*

 

## Statistical analysis

The percentage data transformed into arc-sin, and then all statistical analysis and analysis of variance were performed by SigmaPlot software. The mean value and standard error were also counted (*Liu et al., 2018*).

## RESULTS

### Exogenous BRs promote growth of shoot and adventitious roots induction but restrain the root elongation

To investigate the effects of exogenous BRs, we adopted rooted plantlets for EBL and/or BRZ treatments. The plant growth and phenotype included the relative increased length, fresh weight, shoot and root diameter, and the number of adventitious roots were measured after 3 weeks of treatment (Fig. 1A). The exogenously applied EBL clearly promoted the shoot length but inhibited the root length. The mean increase in the shoot length when treated with EBL was 1.28 cm, which was larger than the control, which had an increase of 1.03 cm; however, the corresponding mean increase in the root length was 3.03 cm, which was less than for the control, which had an increase of 4.56 cm (Fig. 1B). Application of EBL and BRZ simultaneously also promoted the shoot length increases. The exogenous EBL increased the stem diameter, and the other treatment did not significantly alter the stem and root diameters (Fig. 1C). Additionally, the exogenous EBL treatment slightly increased the fresh weights of the shoot and root (Fig. 1D) and increased the number of adventitious roots (Fig. 1E). However, the exogenous BRZ application did not significantly affect the root length, fresh weights of the shoot and root, and number of the adventitious roots of *E. grandis* (Fig. 1). All the changes described above indicated that exogenous BRs could affect the growth of *E. grandis* plantlets, including elongation, thickening, fresh weight, and adventitious roots induction. Exogenous BRs play a dual role in shoot and root elongation, which accelerated the shoot elongation and delayed the root elongation.

### Exogenous BRs promote xylem cell differentiation by anatomical analysis

After the EBL and/or BRZ treatment, we performed an anatomical analysis of the stem and root (Fig. 2). In the stem, the xylem area ratio was calculated. The xylem ratio in the root was calculated by the area of xylem to the area of stele. The xylem area ratio in the stem under the EBL treatment (14.50%) was higher than that in the control (11.23%; Figs. 2A, 2B, and 2I), which suggested that the EBL promoted the expansion of the xylem area in the stem. However, this ratio was slightly lower for BRZ but higher for the EBL+BRZ treatment, with values of 10.44% and 11.96%, respectively (Figs. 2C, 2D, and 2I). We found a similar result in the root, as the ratios of xylem to stele were higher for both the EBL (53.64%) and EBL+BRZ (45.44%) treatments compared with the control (35.32%; Figs. 2E, 2F, 2H, and 2J). These ratios were also higher for the BRZ treatment, with 49.06%, compared with the control (Figs. 2G and 2J).

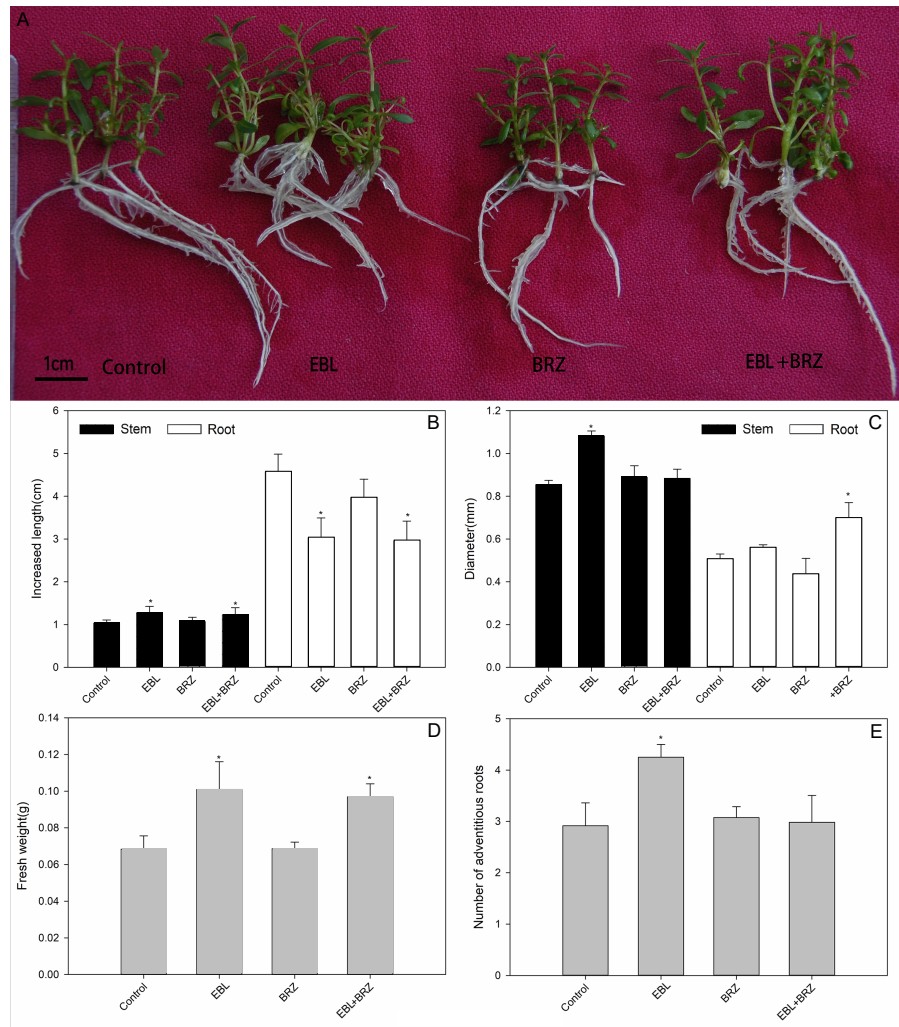

**Figure 1  Effects of different treatment on growth , length, diameter, fresh weight of shoots and roots, and number of roots.** Effects of EBL, BRZ, or EBL + BRZ on growth (A), length (B), diameter (C), fresh weight (D) of shoots and roots, and number of roots (E). The bars represent the standard errors. An asterisk (*) indicates a significant difference from the corresponding value of control.

## Exogenously applied BR regulate the expression of BR biosynthetic and signaling pathway genes

To illustrate the effects of applying EBL on BR metabolism and signal transduction, the expression levels of *EgrDET2*, *EgrCPD*, *EgrDWF4*, *EgrDWF5*, *EgrBZR1*, *EgrBZR2*, *EgrBRI1*, and *EgrBAK1* was determined by using qRT-PCR (Table S1). Most of these genes responded to the exogenously applied EBL treatment (Fig. 3). *EgrCPD*, *EgrDWF4*, *EgrBZR1*, *EgrBZR2*, and *EgrBRI1* were upregulated in the root by the EBL treatment. Correspondingly, *EgrDET2*, *EgrBZR1*, *EgrBZR2*, and *EgrBRI1* were significantly downregulated by the BRZ application. In addition, the EBL+BRZ treatment markedly decreased *EgrCPD* and *EgrDWF5* expression. In the stem, the EBL treatment promoted the expression of *EgrCPD*, *EgrDWF5*, *EgrBZR1*, *EgrBZR2*, and *EgrBRI1,* and they showed downregulation with the BRZ treatment. However,

Peer J

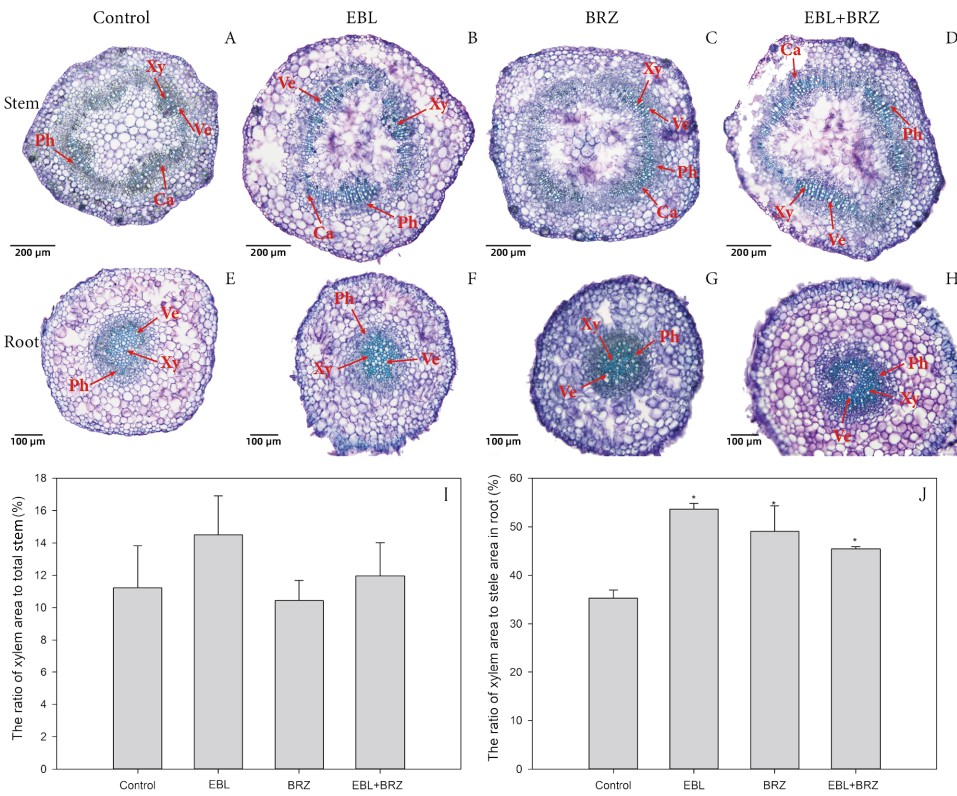

**Figure 2** **Effects of EBL, BRZ, or EBL + BRZ on the anatomical structure of root (A, B, C, and D) and stem (E, F, G, and H), and the xylem area and stem area (I), stele area and root area (J).** (A) (E), (B) (F), (C) (G), and (D) (H) are the cross sections of control, EBL, BRZ, and EBL + BRZ, respectively. Xylem (Xy), cambium (Ca), phloem (Ph), and vascular element (Ve) are attached in photos. I and J respectively represent the ratio of xylem area to the shoot and root. Scale bars represent 200 μm in A, B, C, and D, and 100 μm in E, F, G, and H. The bars represent the standard errors. An asterisk indicates a significant difference from the corresponding value of control in I and J.

EBL+BRZ did not significantly affect their expression levels. The mRNA levels of *EgrDET2*, *EgrCPD*, *EgrBZR1*, *EgrBZR2*, and *EgrBRI1* showed similar expression trends in both the stem and root. However, several genes that responded to the EBL treatment displayed a difference in the stem and root. For example, *EgrDWF5* showed significant increases in the stem but was only slightly altered in the root, whereas *EgrDWF4* was dramatically increased in the root but had almost no changes in the stem (Fig. 3).

## Exogenous BRs regulate the xylem differentiation, lignification, and cellulose synthesis related genes expression

The xylem area increased in the *E. grandis* roots and stems with the EBL treatment. Therefore, several potential genes related to xylem development such as those encoding HD-ZIPIII, MYB, and NAC transcription factors was also applied for analyzing their expression (Fig. 4, Table S2). The exogenously applied EBL dramatically increased *EgrPHV* and *EgrSND1* expression, but significantly inhibited *EgrATHB15* and *EgrMYB83* expression. *EgrVND6*, *EgrATHB15*, and *EgrMYB83* levels significantly decreased, and no

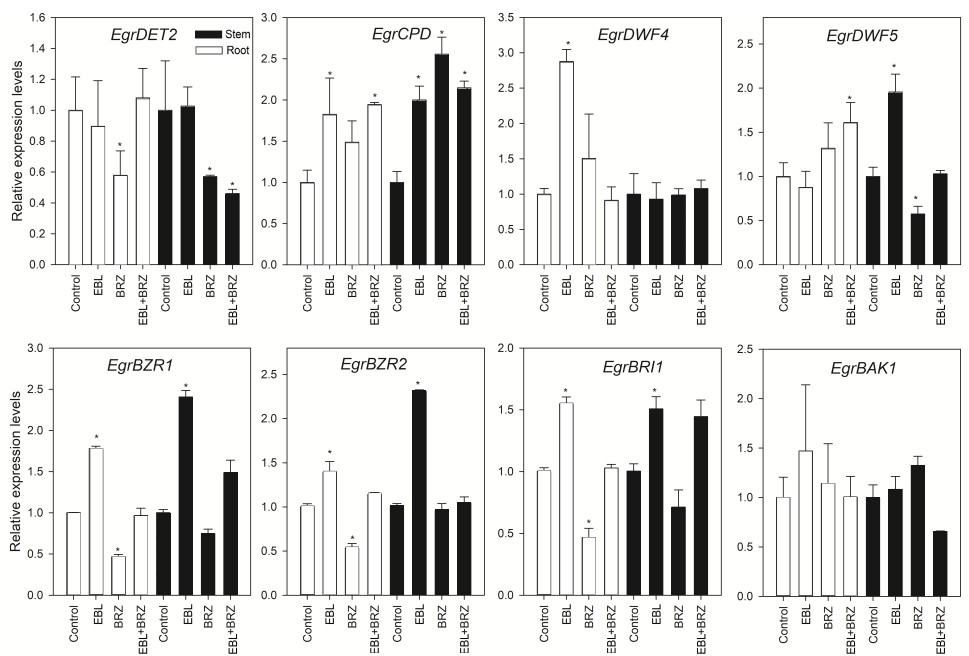

**Figure 3 Expression analysis of genes involved in BR biosynthetic pathway and BR signaling in response to EBL, BRZ, or EBL + BRZ application.** The bars represent the standard errors. An asterisk (*) indicates a significant difference from the corresponding value of control.

gene expressions increased in the stem with the EBL treatment. Similarly, the expression levels of *EgrPHV*, *EgrREV*, *EgrATHB15*, *EgrVND6*, and *EgrMYB83* sharply decreased in the root with the EBL+BRZ treatment. We also performed a CESA genes expression analysis (Fig. 4). The EBL significantly promoted *EgrCESA1* and *EgrCESA3* expression but inhibited *EgrCESA7* mRNA levels in the root. In contrast, the mRNA levels of *EgrCESA1* and *EgrCESA3* were significantly reduced, and *EgrCESA4* and *EgrCESA7* levels increased in the stem with the EBL treatment. However, the BRZ and EBL+BRZ treatments did not affect the expression of most of these four genes. Together, these results suggest that the exogenous BRs application altered the expression levels of the genes involved in xylem development.

## Exogenous BRs promotes elongation and secondary xylem increases in *E. grandis* stem

To verify the effect of the exogenously applied BRs in secondary xylem development in *E. grandis*, EBL was directly applied to the xylem of the stems after debarking 6-month-old plants. The treatment promoted their growth at 1 and 4 weeks. In contrast to the control, the plant height significantly increased with the EBL treatment (Fig. S2). Next, we observed a cross section of the stem that underwent treatment. The amount of newly developed secondary xylem increased in the EBL-treated samples (Figs. 5A and 5B). The number of layers of the newly developing secondary xylem cells of the BR-treated stems (13.25 ± 1.25) was significantly more than the control (7.25 ± 0.37; Table 1). The diameter of

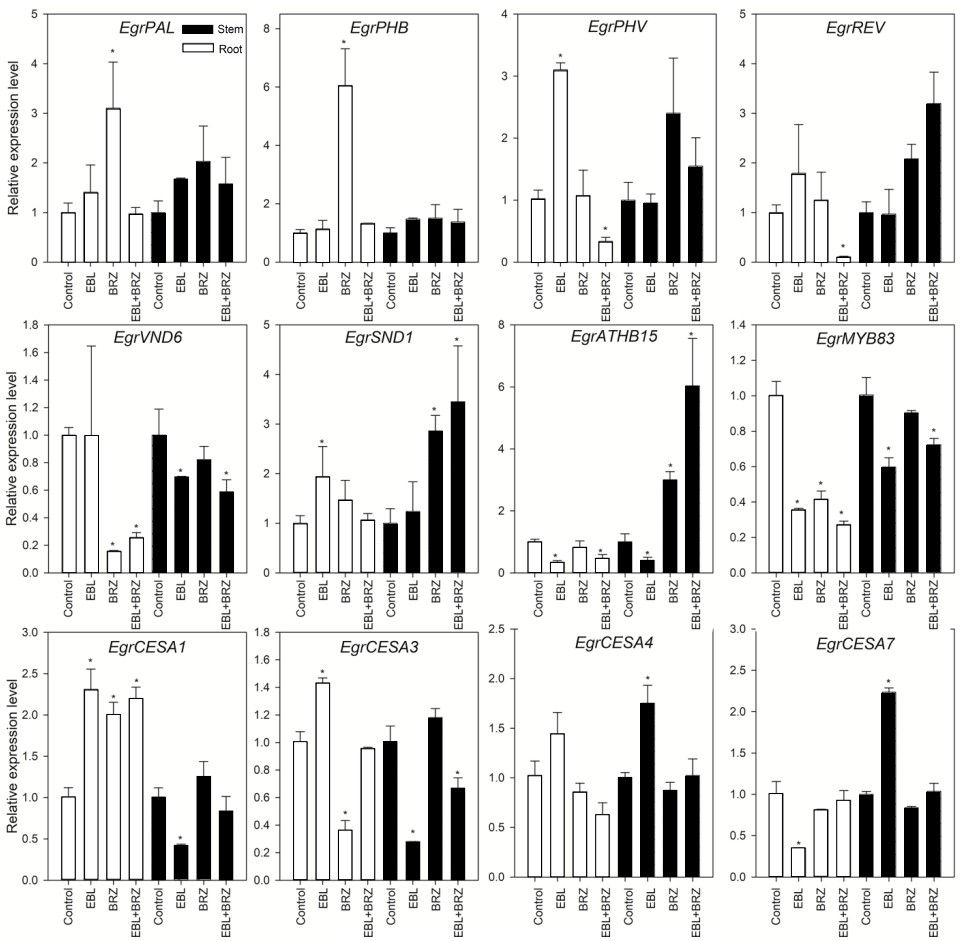

**Figure 4 Expression analysis of genes involved in xylem development in response to EBL, BRZ, or EBL + BRZ application.** The bars represent the standard errors. An asterisk (*) indicates a significant difference from the corresponding value of control.

the fibers and vessel elements in the developing xylem showed significant decreases with the EBL treatment compared with the control after 4 weeks of treatment. A reduction in the number of vessel elements was apparent for the EBL treatment. Additionally, the EBL-treated samples showed increased fiber lengths compared with the controls (Figs. 5C and 5D, Table 1).

To determine the effect of the BR application, BR signaling and secondary cell wall formation-related gene expression was performed (Fig. 6). The BR-signaling key regulators *EgrBZR1*, *EgrBZR2*, and *EgrBRI1* significantly increased at 1 and 4 weeks of treatment (Fig. 6A), suggesting that exogenous BRs application would induce BR-signaling transduction gene expression. The expression of *EgrDWF4* and *EgrDWF5*, which are involved in BR metabolism, significantly decreased at 1 week and then increased at 4 weeks after the EBL treatment. The transcript levels of the secondary cell wall genes *EgrATHB15*, *EgrREV*, *EgrSND1*, and *EgrMYB42* were slightly higher after undergoing BR treatment for 4 weeks,

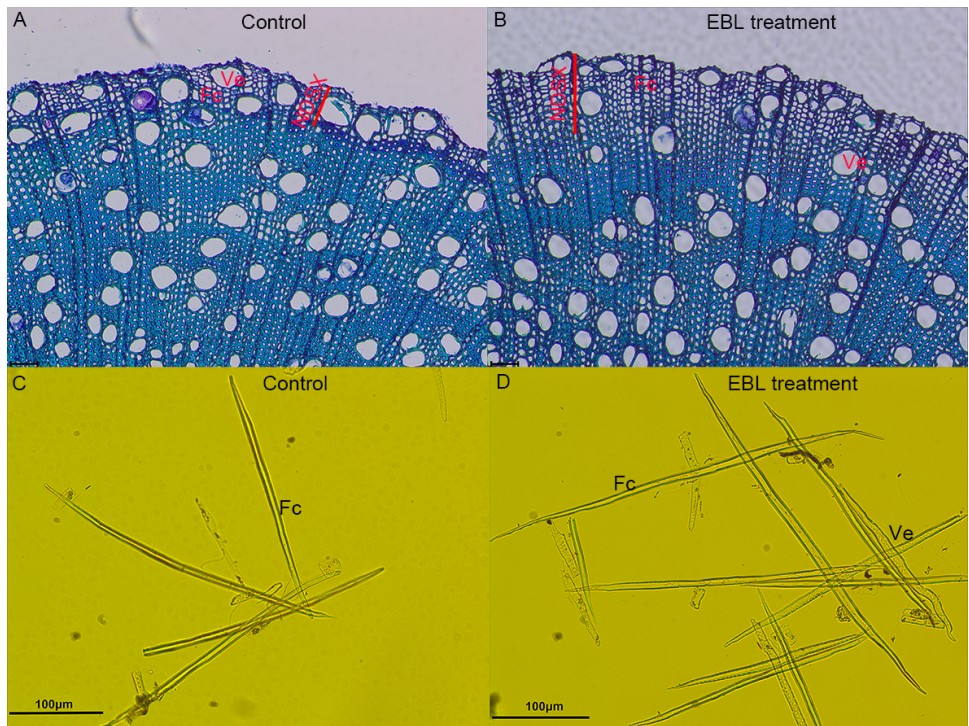

**Figure 5** **Exogenous BR application promoted cell division and fiber cell length.** (A) Cross sectional views of a stem with control and EBL treatment (B), xylem of stem with control (C) and EBL treatment (D). Abbreviations: NDSX, newly developing secondary xylem; Fc, fiber cell; Ve, vessel elements. Scale bars represent 100 μm. Sections in A and B were stained with 0.1% toluidine blue O (TBO). In C&D, the vessel elements or fibers were separated into individual cells using Schulze's reagent.

**Table 1** **Statistical analysis of newly developing secondary xylem cell between EBL-treated plants and corresponding control.**

|  | Layer number of newly developing secondary xylem cell | Fiber cell number per mm$^2$ | Length of fiber (μm) | Number of vessel element | Diameter of vessel element mm |
|---|---|---|---|---|---|
| Control | 7.25 ± 0.38 | 21.50 ± 2.50 | 378.40 ± 11.70 | 4.25 ± 0.75 | 15.15 ± 0.55 |
| EBL treatment | 13.25 ± 1.75[*] | 34.50 ± 2.25[*] | 403.20 ± 17.30[*] | 2.25 ± 0.38[*] | 11.57 ± 0.65[*] |

**Notes.**
Each biological replicate contained at least ten plants for each treatment.
An asterisk (*) indicates a significant difference from the corresponding value of control at $P < 0.05$ level.

whereas those of *EgrREV* and *EgrPHB* were more expressed after undergoing EBL treatment for 1 week. However, the transcription of *EgrPHV*, *EgrMYB83*, *EgrVND6*, and *EgrMYB46* did not increase at 1 and 4 weeks (Fig. 6B). We also investigated the expression levels of *CESA* and lignin synthase genes (Fig. 6C). The exogenous BRs application significantly induced *EgrCESA1* and *EgrCESA3* expression at 1 week and *EgrCESA4* and *EgrCESA7* expression at 4 weeks. However, *EgrHCT* and *EgrC3H* expression was depressed by the EBL treatment at 1 and 4 weeks, respectively; in contrast, *EgrXCP2* and *EgrXCP1* showed an increased expression at 1 and 4 weeks, respectively. All the above results suggest that

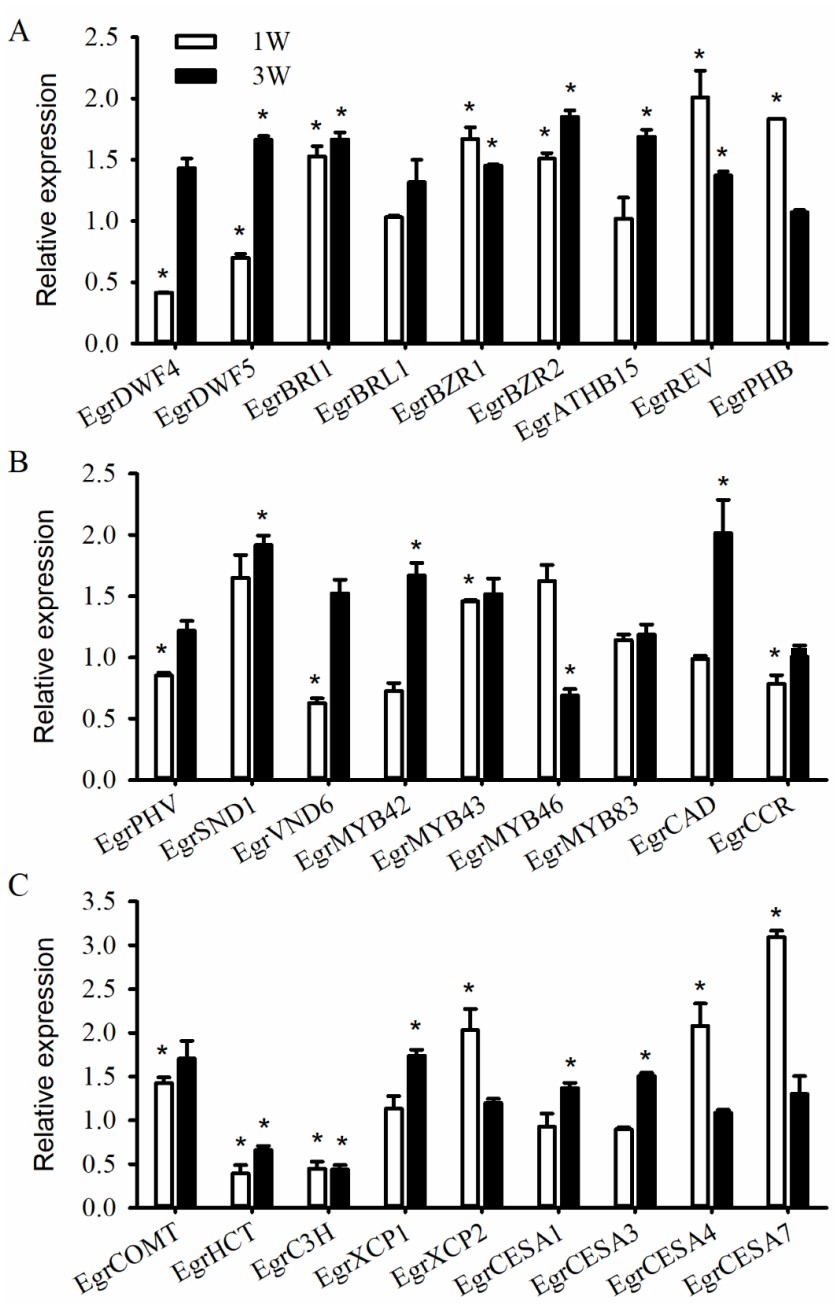

**Figure 6 Expression analysis of genes related to BR signaling.** Expression analysis of genes related to BR signaling (A), secondary xylem development (B), and cellulose and lignin synthesis (C) in response to BR application. The bars represent the standard errors. An asterisk (*) indicates a significant difference from the corresponding value of control.

the exogenously applied BRs altered the secondary xylem development by regulating BR signaling and secondary xylem development-related gene expression.

## DISCUSSION

In addition to providing water and nutrient transport and mechanical support in plants, secondary xylem supplies raw materials for pulp industrial and timber production, and it is used as a sustainable and renewable energy resource. Therefore, complete elucidating of secondary xylem formation, development, and differentiation is particularly important, and consequently, researchers have performed intensive studies on secondary xylem (*Kumar, Campbell & Turner, 2016*; *Mizrachi & Myburg, 2016*). Recently, researchers have recognized and integrated BR into responses to secondary xylem cell differentiation *via* important internal signals (*Du et al., 2020*; *Jiang et al., 2021*; *Lee et al., 2021*). Similarly, they found that an exogenous EBL treatment promoted xylem area increases in the root and stem in *E. grandis*. Moreover, BR (EBL) that was exogenously applied to the vascular cambium resulted in increases in newly developed secondary xylem and fiber cell elongation. In woody plants, BR induced secondary growth and tension wood formation in poplar (*Du et al., 2020*; *Yamagishi et al., 2013*). EBL treatment also increased the diameter and number of fiber cells, promoted cell elongation and accelerated cell division in fiber cells in *L. tulipifera* (*Jin et al., 2014*). Lately, the results of studies where the researchers applied exogenous EBL and BRZ on the xylem tissue of *Leucaena leucocephala* demonstrated that BR could regulate cell wall chemistry components and xylogenesis in woody plants (*Pramod et al., 2021*). Curiously, the diameter of the fiber cells decreased with the BR treatment in the newly developing secondary xylem in *E. grandis*. We propose that this difference resulted from the species responses to the BR concentration. However, we are still unsure why the exogenous EBL and BRZ treatment did not constantly produce contrary results, especially in the root and stem of the shoot from the tissue culture. Researchers have also reported this phenomenon in other species (*Du et al., 2020*; *Pramod et al., 2021*), which suggests that EBL concentration and transportation in plants has an important effect. Hence, we conclude that the plant growth promotion resulted from the fiber cell division accelerating in *E. grandis*. These results demonstrated that BR promotes a xylem increase and plays an essential role in xylem differentiation and development in woody plants.

Researchers have discovered several genes and enzymes that are key for xylem cell patterning and differentiation, including HD-ZIPIIIs, VNDs, and genes for the synthesis of the secondary cell wall main components cellulose and lignin. In *Zinnia elegans*, BR promoted the expression of vascular specification, patterning, and the differentiation of the key HD-ZIPIII gene members *ZeHB10*, *ZeHB11*, and *ZeHB12* (*Ohashi-Ito, Demura & Fukuda, 2002*). In the present study, we induced *EgrATHB8*, *EgrPHB*, and *EgrREV*—which belong to the *HD-ZIPIII* family—and their expression increased with the exogenous application of BR to the xylem of six-month-old plants. However, only *EgrPHV* showed a significantly increased expression with the BR treatment, whereas other members of the *HD-ZIPIII* family, namely *EgrATHB8*, *EgrPHB*, and *EgrREV*, in the stem and root of *E. grandis* showed no corresponding increase or decrease with exogenously applied BR or BRZ. We propose that the difference was due to tissue- and plant-specific behavior. The expression levels of *EgrMYB42*, *EgrMYB43*, and *EgrSND1*—all homologs of the hub genes

regulating xylem vessels and fiber cells differentiation in *Arabidopsis* (*Zhong, Demura & Ye, 2006*; *Zhong et al., 2008*)—dramatically responded to the BR treatment. In addition, *EgrCESA1*, *EgrCESA3*, *EgrCESA4*, and *EgrCESA7* expression showed various increases with the exogenously applied BRs, which occurred with exogenously applied BR in *Arabidopsis* seedlings (*Xie, Yang & Wang, 2011*). Unexpectedly, several cellulose synthase genes showed decreases with the BR treatment in *L. tulipifera* (*Jin et al., 2014*). However, expressions of the key lignin biosynthetic genes *C3H* and *HCT* were reduced by exogenously applying BR in *E. grandis*, which was similar with the result in *L. tulipifera* (*Jin et al., 2014*).

Researchers have extensively studied the signaling networks involved in BR-regulated gene expression (*Belkhadir & Jaillais, 2015*; *Nolan et al., 2020*; *Zhu, Sae-Seaw & Wang, 2013*). The expression of the genes *EgrBRI1*, *EgrBZR1, and EgrBZR2* increased with the BR treatment and decreased with the BRZ treatment in the root and stem of the *E. grandis* plantlets. Their expressions significantly increased when BR was exogenously applied to the xylem of the stem. Similarly, the expression levels of *DcBRI1* and *DcBZR1* in the petioles of carrots and in the lateral roots of *Malus hupehensis* were significantly upregulated with the exogenous BRs treatment (*Mao et al., 2017*; *Que et al., 2017*). The expression of *BRI1*, *BAK1*, and *BES1* and *BZR1* also slightly increased 1 week after BR application in *L. tulipifera* (*Jin et al., 2014*). These results imply that BR-signaling transduction was modestly upregulated with the exogenously applied BRs (*Shi et al., 2022*). Researchers have identified BRI1, which increased vascularization and promoted the expression of the genes involved in secondary cell wall biosynthesis and vascular development, in *Arabidopsis* (*Oh et al., 2011*) and poplar (*Jiang et al., 2021*). They found that BZR1 and BZR2 directly bound to promoter regions of multiple secondary cell wall-related genes such as transcription factors NAC and MYB, cellulose synthase, and lignin synthesis genes (*Jiang et al., 2021*; *Lee et al., 2021*; *Sun et al., 2010*; *Yu et al., 2011*) In addition, it has been found that BR application to BR-deficient tomato mutants restored the normal phenotype of xylem development, while knockdown of *SlBRI1* impaired BR signaling, resulting in xylem failure to differentiate normally and secondary growth with severe defects (*Lee et al., 2019*), and the same results were verified in poplar (*Wang et al., 2022*). Hence, we conclude that BR promotes xylem cell differentiation and development through the *BRI1*-, *BZR1*-, and *BZR2*-regulated signal transduction pathway. However, the regulatory process of BR is also very complex, and there is no more deeply study on the molecular mechanism of *BZR1*, *BZR2* and *BRI1* in secondary xylem development, and further research is needed.

## CONCLUSION

Exogenously applied BRs dramatically increased the shoot length and diameter. Meanwhile the xylem area of the stem and root increased under EBL treatment. During this process, *EgrBRI1*, *EgrBZR1*, and *EgrBZR2* expression were induced while the expressions of HD-ZIPIII and cellulose synthase genes were also altered. Furthermore, the xylem area, fiber cell length, and cell numbers of six-month-old trees under EBL treatment also showed considerable increases. Correspondingly, expression of BR-signaling genes, secondary xylem development-related transcription factor genes also showed significant increases.

It concluded that BR plays an essential role in regulating xylem differentiation and development via the BR-signaling pathway in woody plants.

## ACKNOWLEDGEMENTS

The authors are grateful to the editors and the reviewers for their valuable comments and help. We also thank Dr. Xu Jianmin for giving us some valuable suggestions in manuscript writing and revision.

### Funding

This work was supported by the Natural Science Foundation of Guangdong Province (2020A1515011059) and the National Natural Science Foundation of China (Grant No. 31400554). The funders had no role in study design, data collection and analysis, decision to publish, or preparation of the manuscript.

### Grant Disclosures

The following grant information was disclosed by the authors:
Natural Science Foundation of Guangdong Province: 2020A1515011059.
National Natural Science Foundation of China: 31400554.

### Competing Interests

The authors declare there are no competing interests.

### Author Contributions

- Fangping Zhou performed the experiments, analyzed the data, prepared figures and/or tables, authored or reviewed drafts of the article, and approved the final draft.
- Bing Hu performed the experiments, analyzed the data, prepared figures and/or tables, authored or reviewed drafts of the article, and approved the final draft.
- Juan Li performed the experiments, analyzed the data, prepared figures and/or tables, and approved the final draft.
- Huifang Yan performed the experiments, analyzed the data, prepared figures and/or tables, and approved the final draft.
- Qianyu Liu performed the experiments, analyzed the data, prepared figures and/or tables, and approved the final draft.
- Bingshan Zeng analyzed the data, authored or reviewed drafts of the article, and approved the final draft.
- Chunjie Fan conceived and designed the experiments, analyzed the data, prepared figures and/or tables, authored or reviewed drafts of the article, and approved the final draft.

### Data Availability

The raw measurements are available in the Supplemental File.

## Supplemental Information

Supplemental information for this article can be found online at http://dx.doi.org/10.7717/peerj.16250#supplemental-information.

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
