# Peer review of "Exogenous applications of brassinosteroids promote secondary xylem differentiation in Eucalyptus grandis"

_PeerJ, doi:10.7717/peerj.16250_

## Round 0.1 · original submission · Minor Revisions

Please address our reviewers' comments.

·

Basic reporting

In recent years, there are many reports that BR pathway is involved in cambium cell division, xylem development and secondary wall formation, which can be analyzed in the introduction and discussion。
There are many grammatical errors

Experimental design

1. Due to the use of multiple hormones and inhibitors (24-epibrassinolide (EBL) and/or the BR biosynthesis inhibitor brassinazole (BRZ)) in the manuscript, it should be clear in the manuscript whether each treatment is carried out on different plants or the same plant material is repeatedly processed/This may lead to different experimental results.
2. Figure 2 The slice is not clear .
3. As cytological statistics, how many slices of plant samples have been counted?

Validity of the findings

Brassinosteroids (BRs) is involved in cambium cell division, xylem development and secondary wall formation. In this research, E. grandis plants treated with 24-epibrassinolide (EBL), and/or BR biosynthesis inhibitor brassinazole. The results showed that the BRs dramatically increased the shoot length and diameter, and the exogenous BR increased the xylem area of the stem and root. In this process, several transcription factors related to xylem development were induced. The xylem area, cell length, and cell numbers was considerable increases. Thus, BR had regulatory roles in secondary xylem development and differentiation via the BR-signaling pathway in E. grandis.The result is very interesting,and is consistent with the reported effect of BR pathway.

Additional comments

no comment

Reviewer 2 ·

Basic reporting

Literature references, sufficient context have been provied. The article structure, figures and Tables are profitional. Raw data are shared.

Experimental design

This article The experiment were well organized, and the investigation were performed to a high technical and ethical standard.

Validity of the findings

All underlying data have been provided. Conclusions are well stated, and they are supported by the results.

Additional comments

Zhou et al. treated Eucalyptus grandis plantlets by brassinosteroids to reveal the changes in morphological, histochemical and key genes expression level in root and xylem. The results showed that the BRs applied in this research dramatically increased the shoot length and diameter, and an increased xylem area were also observed in the stem and root. Important transcription factor HD-ZIPIII and cellulose synthase genes that have essential roles in secondary xylem development were altered, and genes involved in BR signaling pathway were also induced.
Here, I have some suggestions which should be revised by the authors.

1.The language should be improved to ensure that an international audience can clearly understand your text. For example, “We grew each tree planted in an individual pot ......”in Line 101. And, passive sentence patterns could appropriately used to replace “We ....” in “Material and methods”.

2.Why the authors chose both stem and root for analysis?

3.What does “xylem area” means? How to calculate “xylem area”

4.In figure 3, the results showed that the relative expression level of the eight genes were the same in control. Do these genes have tissue specificity? If the authors set control as 1 in root and stem separately, double coordinates are required to reflect the differences in gene expression in the two tissues.

5.Line 108:“about 1.0 cm length”should be “about 1.0 cm in length”.

6.Line 277: This sentence should be revised.

7.Line 239-241: This sentence should be revised.

8.Line 246-248: This sentence should be revised.

9.“Table 1” should be added in Line 440, and Header should be added in the table.

10.“peerj-82139-Supplememt_Figure_1” should be “peerj-82139-Supplement_Figure_1”in“supplemental”file.

Reviewer 3 ·

Basic reporting

no comment

Experimental design

no comment

Validity of the findings

no comment

Additional comments

Eucalyptus grandis are one of the most important sources of timber and pulp in the world because they grow rapidly, adapt widely, and have a superior wood property. Further studies indicated that BRs play a critical role in plant growth and development, such as cell division and elongation, vascular differentiation, and cellulose biosynthesis. However, the effects of BRs regarding to secondary xylem development and wood formations in Eucalyptus are still largely unknown. In there, the authors used the 24-epibrassinolide (EBL) to treated the plantlets of Eucalyptus. Then they recorded the plant growth, analyzed the cell structure of the root and stem. And they also performed gene expression analysis, such as secondary growth, BR synthesis and signaling-related genes. Their results showed that the BRs can dramatically increase the shoot length and diameter, and the xylem area of the stem and root. It provides avaluable information in the biological role of BR-mediated secondary xylem development and wood formation in Eucalyptus. But, there are still also some points that have to be addressed for further improvement of this manuscript.
Some minors issues:
1. The genes should be italic, such as Line 433, “EgrBRI1, EgrBZR1, and EgrBZR2” should be written “EgrBRI1, EgrBZR1, and EgrBZR2”
2. The writing of “exogenous BRs”is not uniform in this paper, such as “exogenous BR” or “exogenous BRs”.
3. Format of the citation and paper reference and is inconformity with the PeerJ. Such as the “citation 1” should be written “Shahzad et al., 2018”, the “reference 1” should be written “Shahzad, B.; Tanveer, M.; Che, Z.; Rehman, A.; Cheema, S. A.; Sharma, A.; Song, H.; Rehman, S. u.; Zhaorong, D. 2018. Role of 24-epibrassinolide (EBL) in mediating heavy metal and pesticide induced oxidative stress in plants: A review. Ecotoxicology and Environmental Safety 147: 935-944.”
4. Format of the paper reference should be unified. Such as line 329, the reference have no page number.
5. Line 433, “Developmental cell” should be written “Developmental Cell”.
6. Line 442, “Fiber cell number per mm2” should be written “Fiber cell number per mm2”.

---

## Round 0.2 · Minor Revisions

I am generally satisfied with your responses to our reviewers, but there are still a few issues that required your attention:


A) BRZ inhibits endogenous BR biosynthesis. Why, then, does BRZ application promote shoot length increase? I think this should be discussed.

B) Why was an arc-sin transformation of the data performed, rather than a different one? I wonder how realistic such a transformation is, and how it could be theoretically justified.


C) "Each jar contained five explants, with four jars per replicate" contradicts the legend in Table 1 (" Each biological replicate contained ten plants for each treatment") Please check the tet and explain the discrepancy. In this table, please indicate the units of "diameter of vessel element" and state the size of the sample where the vessel elements were counted (1 mm2? 25 mm2? 100 um2?...)

D) The legend of Figure 2 is wrong: it appears to be a near-copy of the legend of Figure 1

E) In Figures 1, 3, and 4, the bars showing the that the striped bars refer to "stem" and the solid bars refer to "root" are not immediately visible. I suggest that information be also placed on the figure legend

F) The legend to Figure 5 should state exactly what technique is used for coloring/visualizing the structures. It is apparent (from the differences between A and C) that those are not the same, but the reader is left with no information regarding how to reproduce the figures in their own lab.

---

## Round 0.3 · Minor Revisions

The previous Academic Editor is not available so I have taken over handling your submission.

Two reviewers have recommended accepting this article but one reviewer has identified some minor revisions. Please carefully improve the manuscript.

·

Basic reporting

The manuscript reported that promote
secondary xylem diûerentiation in Eucalyptus grandis with BR treated.The exogenous BR increased the xylem area of the stem and root.Several key BR-signaling genes were induced by the BR
treatment.The conclusion is very interesting and has certain value in understanding the mechanism of action of BR in Eucalyptus grandis.

I have a few small questions:

1. All data should be retained to two sificant digits .

2. How many samples were counted? When counting the area of Xylem, where do the slices come from? How many independent materials have been counted?

3. What is the normal area of Xylem in roots and stems in Eucalyptus grandis? What is the age of the eucalyptus trees used in the study? Is the area of about 30-50% Xylem reasonable? The data in Figure 2E-H cannot support this result Fig 2J. Is the data in Figure 2I from the stem or shoot?

4.Need to supplement data to explain how the BR content in plants changes after EBL, BRZ, or EBL + BRZ treatment?

Experimental design

No

Validity of the findings

OK

Additional comments

No

Reviewer 2 ·

Basic reporting

The language is clear and professional.

Experimental design

The Methods was described with sufficient details.

Validity of the findings

All of the underlying data have been provided, and statistical analysis have been performed.

Additional comments

1. “mg.L-1” should be “mg·L-1” in line 106, the same situation was found in Line 110, 114, 121. Please check the whole manuscript .
2.Line 131-132: This sentence should be revised.

Reviewer 3 ·

Basic reporting

no comment

Experimental design

no comment

Validity of the findings

no comment

---

## Round 0.4 · accepted · Accept

The manuscript has been improved and is acceptable for publication.